# Adapting Neural Networks for the Estimation of Treatment Effects

Claudia Shi[1], David M. Blei[1,2], and Victor Veitch[2]

[1]*Department of Computer Science, Columbia Unitversity*
[2]*Department of Statistics, Columbia University*

## Abstract

This paper addresses the use of neural networks for the estimation of treatment effects from observational data. Generally, estimation proceeds in two stages. First, we fit models for the expected outcome and the probability of treatment (propensity score) for each unit. Second, we plug these fitted models into a downstream estimator of the effect. Neural networks are a natural choice for the models in the first step. The question we address is: how can we adapt the design and training of the neural networks used in the first step in order to improve the quality of the final estimate of the treatment effect? We propose two adaptations based on insights from the statistical literature on the estimation of treatment effects. The first is a new architecture, the Dragonnet, that exploits the sufficiency of the propensity score for estimation adjustment. The second is a regularization procedure, targeted regularization, that induces a bias towards models that have non-parametrically optimal asymptotic properties 'out-of-the-box'. Studies on benchmark datasets for causal inference show these adaptations outperform existing methods. Code is available at github.com/claudiashi57/dragonnet.

## 1  Introduction

We consider the estimation of causal effects from observational data. Observational data is often readily available in situations where randomized control trials (RCT) are expensive or impossible. However, causal inference from observational data must address (possible) confounding factors that affect both treatment and outcome. Failure to adjust for confounders can lead to incorrect conclusions. To address this, a practitioner collects covariate information in addition to treatment and outcome status. The causal effect can be identified if the covariates contain all confounding variables. We will work in this 'no hidden confounding' setting throughout the paper. The task we consider is the estimation of the effect of a treatment $T$ (e.g., a patient receives a drug) on an outcome $Y$ (whether they recover) adjusting for covariates $X$ (e.g., illness severity or socioeconomic status).

We consider how to use neural networks to estimate the treatment effect. The estimation of treatment effects proceeds in two stages. First, we fit models for the *conditional outcome* $Q(t,x) = \mathbb{E}[Y \mid t, x]$ and the *propensity score* $g(x) = P(T = 1|x)$. Then, we plug these fitted models into a downstream estimator. The strong predictive performance of neural networks motivates their use for effect estimation [e.g. SJS16; JSS16; Lou+17; AS17; AWS17; SLK18; YJS18; FLM18]. We will use neural networks as models for the conditional outcome and propensity score.

In principle, using neural networks for the conditional outcome and propensity score models is straightforward. We can use a standard net to predict the outcome $Y$ from the treatment and covariates, and another to predict the treatment from the covariates. With a suitable choice of training objective, the trained models will yield consistent estimates of the conditional outcomes and propensity scores. However, neural network research has focused on *predictive* performance. What is

important for causal inference is the quality of the downstream *estimation*. This leads to our main question: how can we modify the design and training of neural networks in order to improve the quality of treatment effect estimation?

We address this question by adapting results from the statistical literature on the estimation of treatment effects. The contributions of this paper are:

1. A neural network architecture—the Dragonnet—based on the sufficiency of the propensity score for causal estimation.
2. A regularization procedure—targeted regularization—based on non-parametric estimation theory.
3. An empirical study of these methods on established benchmark datasets. We find the methods substantially improve estimation quality in comparison to existing neural-network based approaches. This holds even when the methods degrade predictive performance.

**Setup.** For concreteness, we consider the estimation of the average effect of a binary treatment, though the methods apply broadly. The data are generated independently and identically $(Y_i, T_i, X_i) \overset{\text{iid}}{\sim} P$. The average treatment affect (ATE) $\psi$ is

$$\psi = \mathbb{E}[Y \mid \mathrm{do}(T = 1)] - \mathbb{E}[Y \mid \mathrm{do}(T = 0)].$$

The use of Pearl's $\mathrm{do}$ notation indicates that the effect of interest is causal. It corresponds to what happens if we intervene by assigning a new patient the drug. If the observed covariates $X$ include all common causes of the treatment and outcome—i.e., block all backdoor paths—then the causal effect is equal to a parameter of the *observational* distribution $P$,

$$\psi = \mathbb{E}[\mathbb{E}[Y \mid X, T = 1] - \mathbb{E}[Y \mid X, T = 0]]. \tag{1.1}$$

We want to estimate $\psi$ using a finite sample from $P$. Following equation 1.1, an estimator is

$$\hat{\psi}^Q = \frac{1}{n} \sum_i \left[ \hat{Q}(1, x_i) - \hat{Q}(0, x_i) \right], \tag{1.2}$$

where $\hat{Q}$ is an estimate of the *conditional outcome* $Q(t, x) = \mathbb{E}[Y \mid t, x]$. There are also more sophisticated estimators that additionally rely on estimates $\hat{g}$ of the *propensity score* $g(x) = \mathrm{P}(T = 1 \mid x)$; see section 3.

We now state our question of interest plainly. We want to use neural networks to model $Q$ and $g$. How should we adapt the design and training of these networks so that $\hat{\psi}$ is a good estimate of $\psi$?

## 2  Dragonnet

Our starting point is a classic result, [RR83, Thm. 3],

**Theorem 2.1** (Sufficiency of Propensity Score). *If the average treatment effect $\psi$ is identifiable from observational data by adjusting for $X$, i.e., $\psi = \mathbb{E}[\mathbb{E}[Y \mid X, T = 1] - \mathbb{E}[Y \mid X, T = 0]]$, then adjusting for the propensity score also suffices:*

$$\psi = \mathbb{E}[\mathbb{E}[Y \mid g(X), T = 1] - \mathbb{E}[Y \mid g(X), T = 0]]$$

In words: it suffices to adjust for only the information in $X$ that is relevant for predicting the treatment. Consider the parts of $X$ that are relevant for predicting the outcome but not the treatment. Those parts are irrelevant for the estimation of the causal effect, and are effectively noise for the adjustment. As such, we expect conditioning on these parts to hurt finite-sample performance. Instead, we should discard this information.[1] For example, when computing the expected outcome estimator $\hat{\psi}^Q$, (equation 1.2), we should train $\hat{Q}$ to predict $Y$ from only the part of $X$ relevant for $T$, even though this may degrade the *predictive* performance of $\hat{Q}$.

Here is one way to use neural networks to find the relevant parts of $X$. First, train a deep net to predict $T$. Then remove the final (predictive) layer. Finally, use the activation of the remaining net

as features for predicting the outcome. In other contexts (e.g., images) this is a standard procedure [e.g., Gir+14]. The hope is that the first net will distill the covariates into the features relevant for treatment prediction, i.e., relevant to the propensity score $\hat{g}$. Then, conditioning on the features is equivalent to conditioning on the propensity score itself. However, this process is cumbersome. With finite data, estimation errors in the propensity score model $\hat{g}$ may propagate to the conditional outcome model. Ideally, the model itself should choose a tradeoff between predictive accuracy and the propensity-score representation.

This method inspires Dragonnet,[2] a three-headed architecture that provides an end-to-end procedure for predicting propensity score and conditional outcome from covariates and treatment information. See Figure 1. We use a deep net to produce a representation layer $Z(X) \in \mathbb{R}^p$, and then predict both the treatment and outcome from this shared representation. We use 2-hidden layer neural networks for each of the outcome models $\hat{Q}(0, \cdot) : \mathbb{R}^p \to \mathbb{R}$ and $\hat{Q}(1, \cdot) : \mathbb{R}^p \to \mathbb{R}$. In contrast, we use a simple linear map (followed by a sigmoid) for the propensity score model $\hat{g}$. The simple map forces the representation layer to tightly couple to the estimated propensity scores.

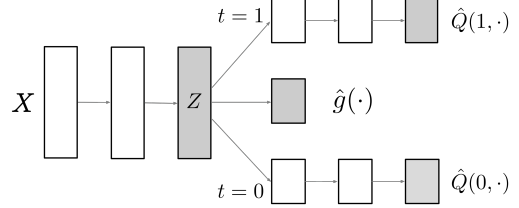

**Figure 1:** Dragonnet architecture.

Dragonnet has parameters $\theta$ and output heads $Q^{\mathrm{nn}}(t_i, x_i; \theta)$ and $g^{\mathrm{nn}}(x_i; \theta)$. We train the model by minimizing an objective function,

$$\hat{\theta} = \operatorname*{argmin}_{\theta} \hat{R}(\theta; \boldsymbol{X}), \quad \text{where} \tag{2.1}$$

$$\hat{R}(\theta; \boldsymbol{X}) = \frac{1}{n} \sum_i \left[ (Q^{\mathrm{nn}}(t_i, x_i; \theta) - y_i)^2 + \alpha \mathrm{CrossEntropy}(g^{\mathrm{nn}}(x_i; \theta), t_i) \right], \tag{2.2}$$

where $\alpha \in \mathbb{R}_+$ is a hyperparameter weighting the loss components. The fitted model is $\hat{Q} = Q^{\mathrm{nn}}(\cdot, \cdot; \hat{\theta})$ and $\hat{g} = g^{\mathrm{nn}}(\cdot; \hat{\theta})$. With the fitted outcome model $\hat{Q}$ in hand, we can estimate the treatment effect with the estimator $\hat{\psi}^Q$ (equation 1.2).

In principle, the end-to-end training and high capacity of Dragonnet might allow it to avoid throwing away any information. In section 5, we study the Dragonnet's behaviour empirically and find evidence that it does indeed trade off prediction quality to achieve a good representation of the propensity score. Further, this trade-off improves ATE estimation even when we use a downstream estimator, such as $\hat{\psi}^Q$, that does not use the estimated propensity scores.

If the propensity-score head is removed from Dragonnet, the resulting architecture is (essentially) the TARNET architecture from Shalit et al. [SJS16]. We compare with TARNET in section 5. We also compare to the multiple-stage method described above.

## 3   Targeted Regularization

We now turn to targeted regularization, a modification to the objective function used for neural network training. This modified objective is based on non-parametric estimation theory. It yields a fitted model that, with a suitable downstream estimator, guarantees desirable asymptotic properties.

We review some necessary results from semi-parametric estimation theory, and then explain targeted regularization. The summary of this section is:

1. $\hat{\psi}$ has good asymptotic properties if it satisfies a certain equation (equation 3.1) with $\hat{Q}$ and $\hat{g}$.
2. Targeted regularization (equation 3.2) is a modification to the training objective.
3. Minimizing this objective forces $(\hat{Q}^{\mathrm{treg}}, \hat{g}, \hat{\psi}^{\mathrm{treg}})$ to satisfy the required equation, where $\hat{Q}^{\mathrm{treg}}$ and $\hat{\psi}^{\mathrm{treg}}$ are particular choices for $\hat{Q}$ and $\hat{\psi}$.

**Setup.** Recall that the general recipe for estimating a treatment effect has two steps: (i) fit models for the conditional outcome $Q$ and the propensity score $g$; (ii) plug the fitted models $\hat{Q}$ and $\hat{g}$ into

a downstream estimator $\hat{\psi}$. The estimator $\hat{\psi}^Q$ in equation 1.2 is the simplest example. There are a wealth of alternatives that, in theory, offer better performance.

Such estimators are studied in the semi-parametric estimation literature; see Kennedy [Ken16] for a readable introduction. We restrict ourselves to the (simpler) fully non-parametric case; i.e., we make no assumptions on the form of the true data generating distribution. For our purposes, the key results from non-parametric theory are of the form: If the tuple $(\hat{Q}, \hat{g}, \hat{\psi})$ satisfies a certain equation, (equation 3.1 below), then, asymptotically, the estimator $\hat{\psi}$ will have various good properties. For instance,

1. robustness in the double machine-learning sense [Che+17a; Che+17b]—$\hat{\psi}$ converges to $\psi$ at a fast rate (in the sample complexity sense) even if $\hat{Q}$ and $\hat{g}$ converge slowly; and
2. efficiency—asymptotically, $\hat{\psi}$ has the lowest variance of any consistent estimator of $\psi$. That is, the estimator $\hat{\psi}$ is asymptotically the most data efficient estimator possible.

These asymptotic guarantees hold if (i) $\hat{Q}$ and $\hat{g}$ are consistent estimators for the conditional outcome and propensity scores, and (ii) the tuple satisfies the *non-parametric estimating equation*,

$$0 = \frac{1}{n} \sum_i \varphi(y_i, t_i, x_i; \hat{Q}, \hat{g}, \hat{\psi}), \tag{3.1}$$

where $\varphi$ is the *efficient influence curve* of $\psi$,

$$\varphi(y, t, x; Q, g, \psi) = Q(1, x) - Q(0, x) + \left( \frac{t}{g(x)} - \frac{1-t}{1-g(x)} \right) \{y - Q(t, x)\} - \psi.$$

See, e.g., Chernozhukov et al. [Che+17b] and van der Laan and Rose [vR11] for details.

A natural way to construct a tuple satisfying the non-parametric estimating equation is to estimate $\hat{Q}$ and $\hat{g}$ in a manner agnostic to the downstream estimation task, and then choose $\hat{\psi}$ so that equation 3.1 is satisfied. This yields the A-IPTW estimator [RRL00; Rob00]. Unfortunately, the presence of $\hat{g}$ in the denominator of some terms can cause the A-IPTW be unstable in finite samples, despite its asymptotic optimality. (In our experiments, the A-IPTW estimator consistently under-performs the naive estimator $\hat{\psi}^Q$.)

Targeted minimum loss estimation (TMLE) [vR11] is an alternative strategy that mitigates the finite-sample instability. The TMLE relies on (task-agnostic) fitted models $\hat{Q}$ and $\hat{g}$. The idea is to perturb the estimate $\hat{Q}$—with perturbation depending on $\hat{g}$—such that the simple estimator $\hat{\psi}^Q$ satisfies the non-parametric estimating equation (equation 3.1). Because the simple estimator is free of $\hat{g}$ in denominators, it is stable with finite data. Thus, the TMLE yields an estimate that has both good asymptotic properties and good finite-sample performance. The ideas that underpin TMLE are the main inspiration for targeted regularization.

**Targeted regularization.** We now describe targeted regularization. We require $Q$ and $g$ to be modeled by a neural network (such as Dragonnet) with output heads $Q^{\mathrm{nn}}(t_i, x_i; \theta)$ and $g^{\mathrm{nn}}(x_i; \theta)$. By default, the neural network is trained by minimizing a differentiable objective function $\hat{R}(\theta; \boldsymbol{X})$, e.g., equation 2.2.

Targeted regularization is a modification to the objective function. We introduce an extra model parameter $\varepsilon$ and a regularization term $\gamma(y, t, x; \theta, \varepsilon)$ defined by

$$\tilde{Q}(t_i, x_i; \theta, \varepsilon) = Q^{\mathrm{nn}}(t_i, x_i; \theta) + \varepsilon \left[ \frac{t_i}{g^{\mathrm{nn}}(x_i; \theta)} - \frac{1 - t_i}{1 - g^{\mathrm{nn}}(x_i; \theta)} \right]$$

$$\gamma(y_i, t_i, x_i; \theta, \varepsilon) = (y_i - \tilde{Q}(t_i, x_i; \theta, \varepsilon))^2.$$

We then train the model by minimizing the modified objective,

$$\hat{\theta}, \hat{\varepsilon} = \underset{\theta, \varepsilon}{\operatorname{argmin}} \left[ \hat{R}(\theta; \boldsymbol{X}) + \beta \frac{1}{n} \sum_i \gamma(y_i, t_i, x_i; \theta, \varepsilon) \right]. \tag{3.2}$$

The variable $\beta \in \mathbb{R}_+$ is a hyperparameter. Next, we define an estimator $\hat{\psi}^{\mathrm{treg}}$ as:

$$\hat{\psi}^{\mathrm{treg}} = \frac{1}{n} \sum_i \hat{Q}^{\mathrm{treg}}(1, x_i) - \hat{Q}^{\mathrm{treg}}(0, x_i), \quad \text{where} \tag{3.3}$$

$$\hat{Q}^{\mathrm{treg}} = \tilde{Q}(\cdot, \cdot; \hat{\theta}, \hat{\varepsilon}). \tag{3.4}$$

The key observation is

$$0 = \partial_\varepsilon \big(\hat{R}(\theta; \boldsymbol{X}) + \beta \frac{1}{n} \sum_i \gamma(y_i, t_i, x_i; \theta, \varepsilon)\big)\big|_{\hat{\varepsilon}} = \beta \frac{1}{n} \sum \varphi(y_i, t_i, x_i; \hat{Q}^{\text{treg}}, \hat{g}, \hat{\psi}^{\text{treg}}). \qquad (3.5)$$

That is, minimizing the targeted regularization term forces $\hat{Q}^{\text{treg}}, \hat{g}, \hat{\psi}^{\text{treg}}$ to satisfy the non-parametric estimating equation equation 3.1.

Accordingly, the estimator $\hat{\psi}^{\text{treg}}$ will have the good non-parametric asymptotic properties so long as $\hat{Q}^{\text{treg}}$ and $\hat{g}$ are consistent. Consistency is plausible—even with the addition of the targeted regularization term—because the model can choose to set $\varepsilon$ to 0, which (essentially) recovers the original training objective. For instance, if $\hat{Q}$ and $\hat{g}$ are consistent in the original model than the targeted regularization estimates will also be consistent. In detail, the targeted regularization model preserves finite VC dimension (we add only 1 parameter), so the limiting model is an argmin of the true (population) risk. The true risk for the targeted regularization loss has a minimum at $\hat{Q} = \mathbb{E}[Y|x,t]$, $\hat{g} = P(T = 1|x)$, and $\hat{\varepsilon} = 0$. This is because the original risk is minimized at these values (by consistency), and the targeted regularization term (a squared error) is minimized at $\hat{Q} + \hat{\varepsilon}H(\hat{g}) = \mathbb{E}[Y|x,t]$, which is achieved at $\hat{\varepsilon} = 0$.

The key idea, equation 3.5, is inspired by TMLE. Like targeted regularization, TMLE introduces an extra model parameter $\varepsilon$. It then chooses $\hat{\varepsilon}$ so that a $\hat{\varepsilon}$-perturbation of $\hat{Q}$ satisfies equation 3.1 with $\hat{\psi}^Q$. However, TMLE uses only the parameter $\varepsilon$ to ensure that the non-parametric estimating equation are satisfied, while targeted regularization adapts the entire model. Both TMLE and targeted regularization are designed to yield an estimate with stable finite-sample behavior and strong asymptotic guarantees. We compare these methods in section 5.

We note that estimators satisfying the non-parametric estimating equation are also 'doubly robust', that is the effect estimate is consistent if either $\hat{Q}$ or $\hat{g}$ is consistent. This property also holds for the targeted regularization estimator, if either the $\hat{Q}$ or $\hat{g}$ is consistent.

# 4 Related Work

The methods connect to different areas in causal inference and estimation theory.

**Representations for causal inference.** Dragonnet is related to papers using representation learning ideas for treatment effect estimation. The Dragonnet architecture resembles TARNET, a two-headed outcome-only model used as the baseline in Shalit et al. [SJS16]. One approach in the literature emphasizes learning a covariate representation that has a balanced distribution across treatment and outcome; e.g., BNNs [JSS16] and CFRNET [SJS16]. Other work combines deep generative models with standard causal identification results. CEVEA [Lou+17], GANITE [YJS18], and CMPGP [AS17] use VAEs, GANs, and multi-task gaussian processes, respectively, to estimate treatment effects. Another approach combines (pre-trained) propensity scores with neural networks; e.g., Propensity Dropout [AWS17] and Perfect Matching [SLK18]. Dragonnet complements these approaches. Exploiting the sufficiency of the propensity score is a distinct approach, and it may be possible to combine it with other strategies.

**Non-parametric estimation and machine learning.** Targeted regularization relates to a body of work combining machine learning methods with semi-parametric estimation theory. As mentioned above, the main inspiration for the method is targeted minimum loss estimation [vR11]. Chernozhukov et al. [Che+17a; Che+17b] develop theory for 'double machine learning', showing that if certain estimating equations are satisfied then treatment estimates will converge at a parametric ($O(1/\sqrt{n})$) rate even if the conditional outcome and propensity models converge much more slowly. Farrell et al. [FLM18] prove that neural networks converge at a fast enough rate to invoke the double machine learning results. This gives theoretical justification for the use of neural networks to model propensity scores and conditional expected outcomes. Targeted regularization is complementary: we rely on the asymptotic results for motivation, and address the finite-sample approach.

# 5 Experiments

Do Dragonnet and targeted regularization improve treatment effect estimation in practice? Dragonnet is a high-capacity model trained end-to-end: does it actually discard information irrelevant to the propensity score? TMLE already offers an approach for balancing asymptotic guarantees with finite sample performance: does targeted regularization improve over this?

We study the methods empirically using two semi-synthetic benchmarking tools. We find that Dragonnet and targeted regularization substantially improve estimation quality. Moreover, we find that Dragonnet exploits propensity score sufficiency, and that targeted regularization improves on TMLE.

**Table 1:** Dragonnet with targeted regularization is state-of-the-art among neural network methods on the IHDP benchmark data. Entries are mean absolute error (and standard error) across simulations. Estimators are computed with the training and validation data ($\Delta_{in}$), heldout data ($\Delta_{out}$), and all data ($\Delta_{all}$). Note that using all the data for both training and estimation improves estimation relative to data splitting. Values from previous work are as reported in the cited papers.

| Method | $\Delta_{in}$ | $\Delta_{out}$ | $\Delta_{all}$ |
|---|---|---|---|
| BNN [JSS16] | $0.37 \pm .03$ | $0.42 \pm .03$ | — |
| TARNET [SJS16] | $0.26 \pm .01$ | $0.28 \pm .01$ | — |
| CFR Wass[SJS16] | $0.25 \pm .01$ | $0.27 \pm .01$ | — |
| CEVAEs [Lou+17] | $0.34 \pm .01$ | $0.46 \pm .02$ | — |
| GANITE [YJS18] | $0.43 \pm .05$ | $0.49 \pm .05$ | — |
| baseline (TARNET) | $0.16 \pm .01$ | $0.21 \pm .01$ | $0.13 \pm .00$ |
| baseline + t-reg | $0.15 \pm .01$ | $0.20 \pm .01$ | $0.12 \pm .00$ |
| Dragonnet | $0.14 \pm .01$ | $0.21 \pm .01$ | $0.12 \pm .00$ |
| Dragonnet + t-reg | $0.14 \pm .01$ | $0.20 \pm .01$ | $0.11 \pm .00$ |

## 5.1 Setup

Ground truth causal effects are rarely available for real-world data. Accordingly, empirical evaluation of causal estimation procedures rely on semi-synthetic data. For the conclusions to be useful, the semi-synthetic data must have good fidelity to the real world. We use two pre-established causal benchmarking tools.

**IHDP.** Hill [Hil11] introduced a semi-synthetic dataset constructed from the Infant Health and Development Program (IHDP). This dataset is based on a randomized experiment investigating the effect of home visits by specialists on future cognitive scores. Following [SJS16], we use 1000 realizations from the NPCI package [Dor16].[3] The data has 747 observations.

**ACIC 2018.** We also use the IBM causal inference benchmarking framework, which was developed for the 2018 Atlantic Causal Inference Conference competition data (ACIC 2018) [Shi+18]. This is a collection of semi-synthetic datasets derived from the linked birth and infant death data (LBIDD) [MA98]. Importantly, the simulation is comprehensive—including 63 distinct data generating process settings—and the data are relatively large. Each competition dataset is a sample from a distinct distribution, which is itself drawn randomly according to the data generating process setting. For each data generating process setting, we randomly pick 3 datasets of size either 5k or 10k.

Some of the datasets have overlap violations. That is, $P(T = 1|x)$ can be very close to 0 or 1 for many values of $x$. Although overlap violations are an important area of study, this is not our focus and the methods of this paper are not expected to be appropriate in this setting. As a simple heuristic, we exclude all datasets where the heldout treatment accuracy for Dragonnet is higher than $90\%$; high classification accuracy indicates a strong separation between the treated and control populations. Subject to this criteria, 101 datasets remain.

**Model and Baseline Settings.** Our main baseline is an implementation of the 2-headed TARNET architecture from Shalit et al. [SJS16]. This model predicts only the outcome, and is equivalent to the Dragonnet architecture with the propensity head removed.

**Table 2:** Dragonnet and targeted regularization improve estimation on average on ACIC 2018. Table entries are mean absolute error over all datasets.

| Method | $\Delta_{all}$ |
|---|---|
| baseline (TARNET) | 1.45 |
| baseline + t-reg | 1.40 |
| Dragonnet | 0.55 |
| Dragonnet + t-reg | 0.35 |

**Table 3:** Dragonnet and targeted regularization improve over the baseline about half the time, but improvement is substantial when it does happen. Error values are mean absolute error on ACIC 2018.

| $\psi^Q$ | $\%_{improve}$ | $\uparrow_{avg}$ | $\downarrow_{avg}$ |
|---|---|---|---|
| baseline: | 0% | 0 | 0 |
| + t-reg | 42% | 0.30 | 0.11 |
| + dragon | 63% | 1.42 | 0.01 |
| + dragon & t-reg | 46% | 2.37 | 0.01 |

For Dragonnet and targeted regularization, we set the hyperparameters $\alpha$ in equation 2.2 and $\beta$ in equation 3.2 to 1. For the targeted regularization baseline, we use TARNET as the outcome model and logistic regression as the propensity score model. We train TARNET and logistic regression jointly using the targeted regularization objective.

For all models, the hidden layer size is 200 for the shared representation layers and 100 for the conditional outcome layers. We train using stochastic gradient descent with momentum. Empirically, the choice of optimizer has a significant impact on estimation performance for the baseline and for Dragonnet and targeted regularization. Among the optimizers we tried, stochastic gradient descent with momentum resulted in the best performance for the baseline.

For IHDP experiments, we follow established practice [e.g. SJS16]. We randomly split the data into test/validation/train with proportion $63/27/10$ and report the in sample and out of sample estimation errors. However, this procedure is not clearly motivated for parameter estimation, so we also report the estimation errors for using all the data for both training and estimation.

For the ACIC 2018 experiments, we re-run each estimation procedure 25 times, use all the data for training and estimation, and report the average estimate errors.

**Estimators and metrics.** For the ACIC experiments, we report mean absolute error of the average treatment effect estimate, $\Delta = \left| \hat{\psi} - \psi \right|$. For IHDP, following established procedure, we report mean absolute difference between the estimate and the sample ATE, $\Delta = \left| \hat{\psi} - 1/n \sum_i Q(1, x_i) - Q(0, x_i) \right|$. By default, we use $\hat{\psi}^Q$ as our estimator, except for models with targeted regularization, where we report $\hat{\psi}^{\text{treg}}$ (equation 3.4). For estimation, we exclude any data point with estimated propensity score outside $[0.01, 0.99]$.

## 5.2 Effect on Treatment Estimation

The IHDP simulation is the de-facto standard benchmark for neural network treatment effect estimation methods. In table 1 we report the estimation error of a number of approaches. Dragonnet with targeted regularization is state-of-the-art among these methods. However, the small sample size and limited simulation settings of IHDP make it difficult to draw conclusions about the methods. The main takeaways of table 1 are: i) Our baseline method is a strong comparator and ii) reusing the same data for fitting the model and computing the estimate works better than data splitting.

The remaining experiments use the Atlantic Causal Inference Conference 2018 competition (ACIC 2018) dataset. In table 2 we report the mean absolute error over the included datasets. The main observation is that Dragonnet improves estimation relative to the baseline (TARNET), and adding targeted regularization to Dragonnet improves estimation further. Additionally, we observe that despite its asymptotically optimal properties, TMLE hurts more than it helps on average. Double robust estimators such as the TMLE are known to be sensitive to violations of assumptions in other contexts [KS07]. We note that targeted regularization can improve performance even where TMLE does not.

In table 2, we report average estimation error across simulations. We see that Dragonnet and targeted regularization improve the baseline estimation. Is this because of small improvement on most datasets or major improvement on a subset of datasets? In table 3 we present an alternative comparison. We divide the datasets according to whether each method improves estimation relative to the baseline. We report the average improvement in positive cases, and the degradation in negative ones. We observe that Dragonnet and targeted regularization help about half the time. When the methods do help, the improvement is substantial. When the methods don't help, the degradation is mild.

## 5.3 Why does Dragonnet work?

Dragonnet was motivated as an end-to-end version of a multi-stage approach. Does the end-to-end network work better? We now compare to the multi-stage procedure, which we call NEDnet.[4] NEDnet has essentially the same architecture as Dragonnet. NEDnet is first trained using a pure treatment prediction objective. The final layer (treatment prediction head) is then removed, and replaced with an outcome-prediction neural network matching the one used by Dragonnet. The representation layers are then frozen, and the outcome-prediction network is trained on the pure outcome prediction task. NEDnet and Dragonnet are compared in table 4. The end-to-end Dragonnet produces more accurate estimates.

**Table 4:** Dragonnet produces more accurate estimates than NEDnet, a multi-stage alternative. Table entries are mean absolute error over all datasets.

| IHDP | $\hat{\psi}^{\mathrm{Q}}$ | $\hat{\psi}^{\mathrm{TMLE}}$ | | ACIC | $\hat{\psi}^{\mathrm{Q}}$ | $\hat{\psi}^{\mathrm{TMLE}}$ |
|---|---|---|---|---|---|---|
| Dragonnet | $0.12 \pm 0.00$ | $0.12 \pm 0.00$ | | Dragonnet | $0.55$ | $1.97$ |
| NEDnet | $0.15 \pm 0.01$ | $0.12 \pm 0.00$ | | NEDnet | $1.49$ | $2.80$ |

We motivated the Dragonnet architecture by the sufficiency of the propensity score for causal adjustment. This architecture improves estimation performance. Is this because it is exploiting the sufficiency? Three observations suggest this is the case.

First, compared to TARNET, Dragonnet has worse performance as a predictor for the outcome, but better performance as an estimator. See Figure 2. This is the case even when we use the simple estimator $\hat{\psi}^Q$, which does not use the output of the propensity-score head of Dragonnet. This suggests that, as intended, the shared representation adapts to the treatment prediction task, at the price of worse predictive performance for the outcome prediction task.

Second, Dragonnet is supposed to predict the outcome from only information relevant to $T$. If this holds, we expect Dragonnet to improve significantly over the baseline when there is a large number of covariates that influence only $Y$ (i.e., not $T$). These covariates are effectively "noise" for the causal estimation since they are irrelevant for confounding. As illustrated in Figure 3, when most of the effect on $Y$ is from confounding variables, the differences between Dragonnet and the baseline are not significant. As the number of covariates that only influence $Y$ increases, Dragonnet becomes a better estimator.

Third, Dragonnet and TARNET should perform equally well with infinite data. With finite data, we expect Dragonnet to be more data efficient as it discards covariates that are irrelevant for confounding. We verify this intuition by comparing models performance with various amount of data. We find Dragonnet's improvement is more drastic with smaller-sized data. See Appendix A for details.

## 5.4 When does targeted regularization work?

The guarantees from non-parametric theory are asymptotic, and apply in regimes where the estimated models closely approximate the true values. We divide the datasets according to the error of the simple ($Q$-only) baseline estimator. As shown in table 6, in cases where the initial estimator is good, TMLE and targeted regularization behave similarly. In cases where the initial estimator is poor, TMLE significantly degrades estimation quality, but targeted regularization does not. It appears that adapting the entire model to satisfy the non-parametric estimating equation avoids some bad finite

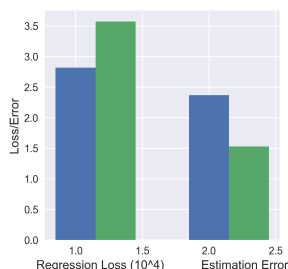

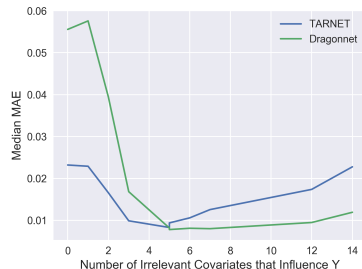

**Figure 2:** Dragonnet has worse prediction loss on the held out data than baseline, but better estimation quality. The estimation error and loss are from a separate run of the ACIC dataset where we held out 30% of data to compute the loss.

**Figure 3:** Dragonnet improves over the baseline if many covariates are irrelevant for treatment. We stratified the ACIC datasets by the number of irrelevant covariates and compared the median MAE across strata.

sample effects. We do not have a satisfactory theoretical explanation for this. Understanding this phenomena is an important direction for future work.

## 6 Discussion

There are a number of directions for future work. Foremost, although TMLE and targeted regularization are conceptually similar, the methods have different performance in our experiments. Understanding the root causes of this behavior may shed insight on the practical use of non-parametric estimation methods. Relatedly, another promising direction is to adapt the well-developed literature on TMLE [e.g., vR11; LG16] to more advanced targeted regularization methods. For instance, there are a number of TMLE approaches to estimating the average treatment effect on the treated (ATT). It is unclear which of these, if any, will yield a good targeted-regularization type procedure. Generally, extending the methods here to other causal estimands and mediation analysis is an important problem.

There are also interesting questions about Dragonnet-type architectures. We motivated Dragonnet with the intuition that we should use only the covariate information that is relevant to both the treatment assignment and outcome. Our empirical results support this intuition. However, in other contexts, this intuition breaks down. For example, in RCT, where covariates only affect the outcome, adjustment can increase power [SSI13]. This is because adjustment may reduce the effective variance of the outcome, and double robust methods can be used to ensure consistency (the treatment assignment model is known trivially). Our motivating intuition is well supported in the large-data, unknown propensity-score model case we consider. It would be valuable to have a clear articulation of the trade-offs involved and practical guidelines for choosing covariates for causal adjustment. As an illustration, recent papers have used Dragonnet-type models for causal adjustment with black-box embedding methods [VWB19; VSB19]. They achieve good estimation accuracy, but it remains unclear exactly what trade-offs may being made.

In a different direction, our experiments do not support the routine use of data-splitting in effect estimation. Existing methods have commonly split the data into train and test sets and used predictions on the test set to compute the downstream estimator. This technique has some theoretical justification [Che+17b] (in a $K$-fold variant), but significantly degrades performance in our experiments. We note that this is also true in our preliminary (unreported) experiments with $K$-fold data splitting. A clearer understanding of why and when data splitting is appropriate would be high impact. We note that Farrell et al. [FLM18] prove that data reuse does not invalidate estimation when using neural networks.

## Acknowledgements

We are thankful to Yixin Wang, Dhanya Sridhar, Jackson Loper, Roy Adams, and Shira Mitchell for helpful comments and discussions. This work was supported by ONR N00014-15-1-2209, ONR 133691-5102004 , NIH 5100481-5500001084, NSF CCF-1740833, FA 8750-14-2-0009, the Alfred

P. Sloan Foundation, the John Simon Guggenheim Foundation, Facebook, Amazon, IBM, and the government of Canada through NSERC. The GPUs used for this research were donated by the NVIDIA Corporation.

## Footnotes

[1] A caveat: this intuition applies to the complexity of learning the outcome model. In the case of high-variance outcome and small sample size modeling all covariates may help as a variance reduction technique.

[2]"Dragonnet" because the dragon has three heads.

[3]There is a typo in Shalit et al. [SJS16]. They use setting A of the NPCI package, which corresponds to setting B in Hill [Hil11]

[4]"NEDnet" because the network is beheaded after the first stage.

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
