[Reviews · NeurIPS 2019]

Reviewer 1



The authors present a methodology for a deep learning model for the nuisance parameters for causal estimation (mean outcome and propensity score functions), which is based around learning a shared low-dimensional representation of the confounders regularized for good finite-sample performance, and a learning algorithm for training this model based on their presented concept of "targeted regularization" (which is a regularization scheme inspire by TMLE). Using their novel methodologies they are able to achieve state of the art performance on standard datasets for high-dimensional causal inference. Their methodology combines multiple different ideas in causal inference (multi-headed deep learning models and targeted learning) in a novel way, and their empirical evaluation seems very strong, so I would recommend the paper for acceptance. Some issues with the paper are as follows: - They claim that their methodology is stable because it does not involve any propensity terms in denominators. However as far as I can understand this is false because their prediction is based on their learnt \tilde{Q} function, which involves propensities in denominators in its second term (which is weighted by \hat{epsilon}. - The baselines in their evaluations are not completely clear. In particular it seems like the "baseline (TARNET)" method should be the same as "TARNET (Sha+16)" in Table 1, however they report different numbers in each. It seems like maybe the second is the number reported in past work, and the first is using the author's code possibly with different details in exact model architecture and learning hyperparameters, but this is not made explicit. - The authors claim that part of strength of model is insensitivity to very low/high propensity scores due to lack of propensity scores in denominators. However in their evaluations they exclude data with extreme propensity scores which makes this claim difficult to verify. In addition since different methods will estimate propensity scores differently it is not made clear what data points are removed, and whether different methods are being evaluated on the same data - It seems weird that Equation 2.2 has no hyperparameter for how much the two loss terms are weighted, is there a reason why no such term is included? - They claim that the third head in their model regularizes the model such that finite-sample performance should be improved. However no part of their experiments evaluates this claim. It would be good to see an experiment, even if it's just with synthetic data, that tests this (could be done by using a synthetic data distribution and comparing model performance with 2 vs 3 heads with small n versus very large n)

Reviewer 2



The draft is basically well written, except that Section 3 seems a bit disorganized. Estimation of treatment effects from observational data is an important topic in causal inference. A line of research has been done in recent years. This work uses neural networks for the estimation of treatment effects from observational data in “no hidden confounding” setting. They proposed two methods in terms of two stages. First, they proposed Dragonnet, a three-headed architecture that provides an end-to-end procedure for predicting propensity score and conditional outcome from covariates and treatment information. If the propensity-score head is removed from Dragonnet, the resulting architecture is the TARNET architecture from Shalit et al. Second, they made a modification to the objective function in training. The main inspiration here is the targeted minimum loss estimation (TMLE). My main concerns are the novelty. Both of the methods seem like we have A and B, so we can try to combine to see how it works.

Reviewer 3



Summary: In this paper, the authors address the problem of estimating treatment effects from observational data when all covariates are measured (the ‘no-confounding’ assumption). The estimation proceeds in two stages: in the first step a model for the expected outcome, i.e., Q(t, x) = E[Y | t, x], and one for the propensity score, i.e., g(x) = p(T = 1 | x), are fitted; in the second stage, the average treatment effect is derived from the previously computed fits of Q(t, x) and g(x). The authors focus on improving the models estimated in the first stage with the ultimate goal of improving the treatment effect estimation in the second stage. For this purpose, they propose a neural network architecture called Dragonnet, in which the outcome models Q(0, x) and Q(1, x) are tightly coupled with the propensity score g(x). The authors then propose a procedure called targeted regularization for improving the asymptotic properties of the neural network-estimated functions in terms of estimating the average treatment effect, at the expense of predictive performance. Detailed comments: • Originality: The two methodological contributions appear to be original. Relevant related work is adequately cited in a separate section on page 5, but more references could be added. For instance, the paper “Representation Learning for Treatment Effect Estimation from Observational Data” by Yao et al. is probably highly relevant for this work. • Quality: The ideas proposed appear sound, but are validated only through a limited number of experiments. I have doubts that asymptotic properties like double-robustness are achieved so easily for targeted regularization. It seems that coupling the estimators for Q and g will in general lead to loss of consistency and of the double-robustness property. The authors claim that “consistency is plausible – even with the addition of the targeted regularization term” because “the model can choose to set epsilon to zero”. However, the non-parametric estimating equation, which is needed to achieve the good asymptotic properties, is satisfied only for the value of epsilon that minimizes the modified objective (locally), and this value will not be zero in general. I also did not find the empirical study particularly convincing. For instance, the authors fail to explain how they combined targeted regularization with TMLE in for the experiments described in Table 2 (page 7) and Table 4 (page 8). • Clarity: The submission is well-structured and easy to read for the most part. However, most of the figure and table captions are extremely bare and are not self-contained. What’s more, The authors tend to overuse hedge words and rhetorical questions in their argumentation. • Significance: The two methodological contributions, Dragonnet and targeted regularization, improve on state-of-the-art approaches like TARNET and TMLE, respectively, but only incrementally. It is hard to accurately evaluate how significant these contributions are based on the limited number of experiments. A theoretical analysis of the newly-proposed estimators couple with a more comprehensive experimental section would go a long way towards shedding light on the significance of these ideas. • Minor comments: • The legend is missing in Figure 2. The x-axis scale should be removed. • Is equation (3.5) missing a factor of (-2) on the right side? • On page 7, the authors claim that “targeted regularization essentially never hurts”, yet in Table 3 the targeted regularization degrades the performance of the simple baseline estimator (row 1) in half of the cases. Could there be a typo in the table? • Have the authors verified that “in cases where the targeted regularization loss term is large” the model responds “by setting the parameter epsilon to 0 and recovering the baseline”? • The terms ‘semi-parametric’ and ‘non-parametric’ are used throughout the paper as if interchangeable. For example, at the beginning of Section 3 (page 3): “This modified objective is based on non-parametric estimation theory.” and later “We review some necessary results from semi-parametric estimation theory”. • The footnote explanation for calling the architecture “Dragonnet” is curious as dragons typically have just one head (country to hydras).

[Author Response · NeurIPS 2019]

**Reviewer 1** *[The] methodology combines multiple different ideas in causal inference (multi-headed deep learning models and targeted learning) in a novel way, and their empirical evaluation seems very strong.* Thank you for your support and your insightful remarks and suggestions.

*claim that their methodology is stable because it does not involve any propensity terms in denominators. However [...] their learnt $\tilde{Q}$ function, which involves propensities in denominators in its second term (which is weighted by $\hat{\epsilon}$.* This is a good point. The idea is that the model can use the $\epsilon$ term and the loss term to downweight the influence of extreme propensities. In the case $\hat{\epsilon} = 0$ the propensity score is not used for the estimate. We have clarified the language, and emphasized that insensitivity to extreme propensity-score values is an intuition.

*The baselines in their evaluations are not completely clear.* We have clarified this. We chose hyperparameters (mainly, the optimizer) to produce a stronger baseline version of TARnet (lns 231-235).

*The authors claim that part of strength of model is insensitivity to very low/high propensity scores due to lack of propensity scores in denominators. However in their evaluations they exclude data with extreme propensity scores which makes this claim difficult to verify. In addition [...]* We have clarified this. We used a standard trimming procedure (e.g., as in the expts of van der Laan and Rose). We have also added a table to the appendix comparing trimmed and untrimmed estimators—treg is substantially less affected than AIPTW/TMLE.

*It seems weird that Equation 2.2 has no hyperparameter...* We have clarified this. Indeed, there is a hyperparameter. We used an arbitrary fixed value (1.0) to avoid unfairly advantaging our method via hyperparam search.

*claim that the third head in their model regularizes the model such that finite-sample performance should be improved..."* What we meant is that there is no infinite-data advantage to this procedure. It's not clear what happens in small vs. moderately-sized data. We have clarified the language. We have added the suggested experiment to the appendix. We subsampled the ACIC data. It shows that the Dragonnet's improvement is more significant with smaller-sized data.

**Reviewer 2** *My main concerns are the novelty. Both of the methods seem like we have A and B, so we can try to combine to see how it works.* Both proposed methods are new and non-trivial. R1 and R3 both agree the paper is novel.

**Reviewer 3** Thank you for your comments. We've clarified where you requested. We address your main concerns:
**Experiments** We are deeply surprised by the dismissal of the experiments as 'limited'. The range and detail of experiments go well beyond the level typical for this area (e.g., the related work). The paper makes a good-faith attempt to test using (necessarily) semi-synthetic data that is (i) realistic, (ii) defined by an existing benchmark, and (iii) covers a wide range of simulation approaches (we use 101 ACIC benchmark simulated datasets). Further, we produce a strong baseline (improving on the published version of TARnet) and do not engage in any unfair hyperparameter tuning.

As suggested, we have made various clarifications to the captions of the tables and figures in experiments. We've also clarified the following: TMLE is combined with treg by plugging the $\hat{Q}$ and $\hat{g}$ values from targeted regularization into the TMLE. Figure 2 and 3 are out-of-sample with ACIC data. In 5.4, we divide according to absolute ATE estimation error for the baseline (<1 or >1, chosen arbitrarily as a small value). We've also changed notation away from "$\epsilon$", which is overloaded. Our comment "targeted regularization essentially never hurts" refers to the dragonnet/treg combo, where degradation is very small (0.01) on average. Note table 3 gives improvement/degradation amount conditioned on improvement/degradation. Note, however, **there is a typo in table 3**. We miscopied some numbers; the +dragon values should be 54%, 1.42, 0.32. Note this is better than the incorrectly reported values. We apologize for the error.

**Theory** Our purpose here is to produce practical methods *inspired* by existing theory results. As we note in the paper, consistency of the neural networks is a key ingredient for asymptotic results. You correctly point out that we do not attempt to prove consistency. However, establishing such consistency for neural networks is an active area of research. Extending such results is a valuable direction for future work, but is outside the scope of the paper. We note that the theory inspiring Dragonnet is non-asymptotic and we do not make any claims about Dragonnet's asymptotic properties.

With respect to targeted regularization, you write, "*It seems that coupling the estimators for Q and g will in general lead to loss of consistency and of the double-robustness property. The authors claim that "consistency is plausible – even with the addition of the targeted regularization term" because "the model can choose to set epsilon to zero".*" Our point here is that if $Q$ and $g$ models are consistent without the targeted regularization, then they will also be consistent with it (thus, the conditions for good asymptotics are satisfied). The reason is that treg embeds the original $(Q, g)$ model in a larger model class by introducing $\epsilon$ (with $\epsilon = 0$ the original model class). In detail, consistency in the original model means $\hat{Q} = \mathbb{E}[Y|x,t]$ and $\hat{g} = P(T = 1|x)$ at $n = \infty$. The treg model preserves finite VC dimension (we add only 1 param), so the limiting model is an argmin of the true (population) risk. The true risk for the treg loss has a minimum at $\hat{Q} = \mathbb{E}[Y|x,t]$, $\hat{g} = P(T = 1|x)$, and $\hat{\epsilon} = 0$. This is because the original risk is minimized at these values (by consistency), and the treg term (a squared error) is minimized at $\hat{Q} + \hat{\epsilon}H(\hat{g}) = \mathbb{E}[Y|x,t]$, which is achieved at $\hat{\epsilon} = 0$. This gives efficiency under the same consistency condition as TMLE/AIPTW. We have clarified this point in the exposition and added the proof. Note that there *is* a trade-off in the (finite-data) empirical risk objective or in the presence of model-misspecification (indeed, that's the point).

[Meta-Review · NeurIPS 2019]

The paper addresses the problem of inferring causal effects using observational data, under the “no-hidden confounders” scenario. Recently there has been much interest in the problem from the machine learning community, including several papers proposing neural net architectures tailored for this problem. This paper proposes a new regularization scheme for this task. The idea is inspired by TMLE, a well known method for doubly-robust estimation of treatment effects. However, TMLE is only an inspiration - the regularization scheme and resulting architecture are distinct and novel. Indeed, this is the first time I’ve seen the approach of estimating equations, common in the econometrics literature, being directly and fruitfully taken up to create a novel deep net architecture and optimization objective. The idea is followed by an extensive set of experiments. First, it is shown that the method achieves good results on the widely used IHDP benchmark. Then, the paper uses the large ACIC 2018 benchmark, which includes 101 different and purposely diverse datasets. The proposed method shows excellent results compared to relevant baselines. Then, the paper includes a quite in-depth (considering the space constraints) examination of what drives the method’s performance, using an ablation study and a careful simulation study. Two of the reviewers agreed that the paper has substantial novelty to it. The third reviewer thought it more incremental. One of the reviewers who believed the paper is original and novel was nonetheless concerned about many specific issues relating to clarity and presentation, and also about the depth of the experimental results. That reviewer also had doubts about part of the theoretical results as they pertain to consistency. In my estimation, and after considering the authors response: 1. The idea is definitely novel. Indeed, I find this to be one of the most novel ideas I’ve seen in the last 3 years in the field of applying ML to causal effect inference. 2. The experiments are extensive, and definitely on par with the existing published literature in the field. The subsections “Why does Dragonnet work?” and “When does targeted regularization work?” are exactly the kind of examination I believe we would all want to see in papers proposing novel optimization schemes and architectures. 3. I found the paper to be clear overall. The specific points correctly raised by the reviewers were satisfactorily addressed in the authors response. I believe the point raised by one of the reviewers about consistency stems from a misunderstanding and is explained in the paper itself and the authors response. As the reviewers pointed out, the paper would have gained from a more thorough analysis of this specific regularization helps in the finite-sample case - how is the hypothesis class implied by the regularizer better for causal effect inference? Though this would no-doubt help, I do not think it is a necessary requirement for acceptance. There is enough novelty in the idea in itself. Some other improvements I would like to see are understanding what epsilons are actually chosen by the optimization procedure, and how other baselines perform on the ACIC benchmark.